Journal of Data-centric Machine Learning Research (2026)    Submitted 03/26; Revised 06/26; Published 06/26

# CycliST: A Video Language Model Benchmark for Reasoning on Cyclical State Transitions

**Simon Kohaut**[1,2,*]                                 SIMON.KOHAUT@CS.TU-DARMSTADT.DE

**Daniel Ochs**[1,*]                                      DANIEL.OCHS@CS.TU-DARMSTADT.DE

**Shun Zhang**[1]                                     SHUN.ZHANG@STUD.TU-DARMSTADT.DE

**Benedict Flade**[3]                                        BENEDICT.FLADE@HONDA-RI.DE

**Julian Eggert**[3]                                          JULIAN.EGGERT@HONDA-RI.DE

**Kristian Kersting**[1,5,6,7]                                    KERSTING@CS.TU-DARMSTADT.DE

**Devendra Singh Dhami**[4]                                              D.S.DHAMI@TUE.NL

[1] *Artificial Intelligence and Machine Learning Lab, TU Darmstadt*

[2] *Konrad Zuse School of Excellence in Learning and Intelligent Systems (ELIZA)*

[3] *Honda Research Institute Europe GmbH, Offenbach, Germany*

[4] *Uncertainty in Artificial Intelligence Group, TU Eindhoven*

[5] *Hessian Center for AI (hessian.AI)*

[6] *Center for Cognitive Science*

[7] *German Center for Artificial Intelligence (DFKI)*

[*] *Authors contributed equally*

**Reviewed on OpenReview:** *https: // openreview. net/ forum? id= l03g53HUL2*

**Editor:** Lijie Hu

## Abstract

We present CycliST, a novel benchmark dataset designed to evaluate Video Language Models (VLM) on their reasoning over cyclical state transitions. CycliST captures fundamental aspects of real-world processes in synthetic, richly structured video sequences featuring periodic visual patterns. Furthermore, CycliST offers a tiered evaluation, providing increasing difficulty levels by varying the number of cyclic objects, scene clutter, and lighting conditions. We conduct extensive experiments with current open-source and proprietary state-of-the-art VLMs and reveal their limitations in generalizing to cyclical dynamics, such as linear and orbital motion, as well as to time-dependent changes in visual attributes like color and scale. Our results demonstrate that present-day VLMs struggle to reliably detect and exploit cyclic patterns, lack a notion of temporal understanding, and are unable to extract quantitative insights from scenes, such as the number of objects in motion, highlighting a significant technical gap that needs to be addressed. More specifically, we find no single model consistently outperforms others: neither size nor architecture correlates strongly with outcomes, and no model performs equally well across all tasks. By providing a targeted challenge and a comprehensive evaluation framework, CycliST paves the way for visual reasoning models that surpass the state-of-the-art in understanding periodic patterns.

**Keywords:**  Video Question Answering, Scene Understanding, Spatio-Temporal Reasoning, Benchmark Dataset

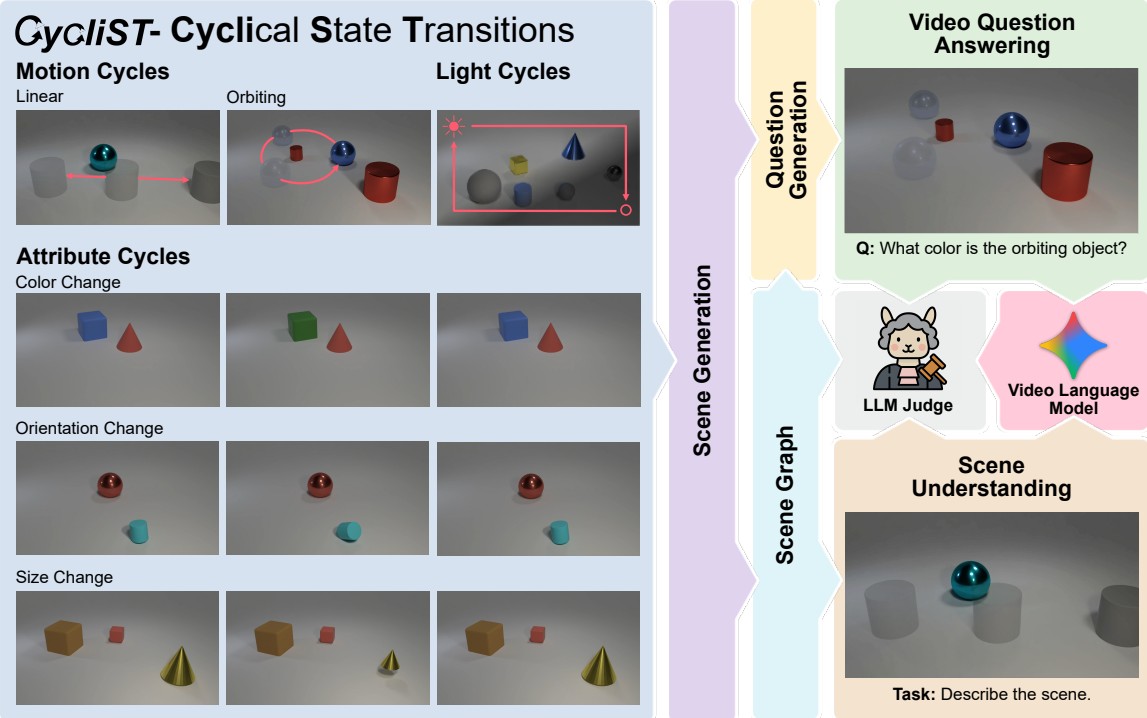

Figure 1: **CycliST**: A diagnostic Video Question Answering and Scene Understanding benchmark for Video Language Models. As CycliST's scenes undergo periodic, smooth visual changes, they always return to each configuration at regular intervals.

## 1 Introduction

Cyclical patterns are a common feature of our environment, and recognizing and interpreting them is fundamental for future Artificial Intelligence models to match human cognition.

From traffic lights on the daily commute to the orbital motion of satellites, repetition is everywhere around us. These recurring phenomena are crucial in many fields. For example, recognizing the timing of traffic lights or the arrival of other vehicles at the intersection is essential for safe navigation of autonomous vehicles (Gautam and Kumar, 2023). Similarly, by measuring the orbital period of the moonlet Dimorphos around the asteroid Didymos before and after the DART impact, scientists were able to precisely quantify its deflection (Johnson et al., 2023). In industrial environments, packages on conveyor belts exhibit cyclical patterns that require monitoring to detect and treat anomalies, such as incorrect timing or variations in package sizes. In healthcare, cyclical patterns, such as heartbeats, are analyzed through medical imaging to detect abnormalities (Gupta et al., 2022).

By understanding and analyzing these cyclical patterns, we can, to some extent, save computational resources by compressing this repetitive information. That is, with a reliable model of a cyclical pattern, an agent may allocate fewer resources to monitoring the evolution of the respective state. Although cyclical phenomena are widespread, current diagnostic datasets primarily cover a range of video contexts but often miss the importance of cyclical and temporal structures embedded in these scenes.

This omission limits their effectiveness in scenarios that require an understanding of these patterns, as they lack the complexity of real-world cyclical events. In contrast, our work aims to address this gap by introducing a dataset specifically designed to highlight cyclical and temporal structures, enhancing the accuracy of reasoning over cyclical videos.

We introduce CycliST (Fig. 1), a benchmark dataset designed to tackle the problem of recognizing and interpreting cyclical scenes from a novel perspective. Inspired by synthetic diagnostic datasets such as CLEVR (Johnson et al., 2017; Yi et al., 2020; Girdhar and Ramanan, 2020; Li et al., 2023), we propose a challenge that focuses on the intricate cyclical and temporal structures of objects. Our dataset is founded on two principles: First, it emphasizes reasoning within cyclical and temporal contexts while maintaining simplicity in its generated scenes and language. Second, CycliST provides comprehensive annotations to support complex reasoning tasks and offer robust diagnostics for model evaluation.

CycliST includes precise ground truth motion paths and cyclical state transitions for each object in the videos. This detailed annotation enables the assessment of AI models in conditions that closely resemble real-world scenarios, providing a robust framework for identifying strengths and weaknesses in current AI technologies. By providing a comprehensive dataset focused on cyclical phenomena, we seek to bridge the gap in current AI research and lay a foundation for future developments in video reasoning.

In summary, we make the following contributions to Video Question Answering:

1. We propose CycliST, a novel high-resolution, high-framerate VLM benchmark for visual reasoning on synthetic scenes of cyclical state transitions.

2. CycliST extends previous work by introducing linear and orbital motion patterns as well as periodic attribute dynamics, including orientation, scale, and color changes.

3. With CycliST, we provide a large diagnostic benchmark (14.8k videos, 120k question-answer pairs), showing how state-of-the-art VLMs are brought to their limits in understanding complex, cyclical scenes, and pointing towards important skills the next VLM generation ought to obtain.

4. We make CycliST, including the generated videos and questions* as well as its Blender-based render pipeline, question generation, and scripts for the presented dataset splits[†], available as open-source repositories.

We proceed as follows. In Section 2, we lay out the fundamental principles employed to generate our benchmark dataset, covering CycliST's time-dynamic object descriptions, motion cycles, attribute cycles, lighting cycles, validation, and rendering techniques. Then, in Section 3, we disclose CycliST's approach to template-based question generation, describing both temporal descriptive and scene representative tasks. Next, Section 4 outlines how CycliST is not only split into train, test, and validation splits, but also how it enables a tiered evaluation of VLMs through fine-grained control of the scenes' complexities. We then provide an experimental evaluation of state-of-the-art VLMs in Section 5, demonstrating how both open-source and proprietary models fail to reliably recognize and describe CycliST's scenes. Following a discussion of related work in Section 6, we conclude with a summary and discussion of limitations and future work in Section 7.

---

*Dataset: https://huggingface.co/datasets/AIML-TUDA/CycliST

[†]Code: https://www.github.com/simon-kohaut/CycliST

| Motion Cycles | | Attribute Cycles | | |
|---|---|---|---|---|
| Linear | Orbit | Size | Color | Orientation |

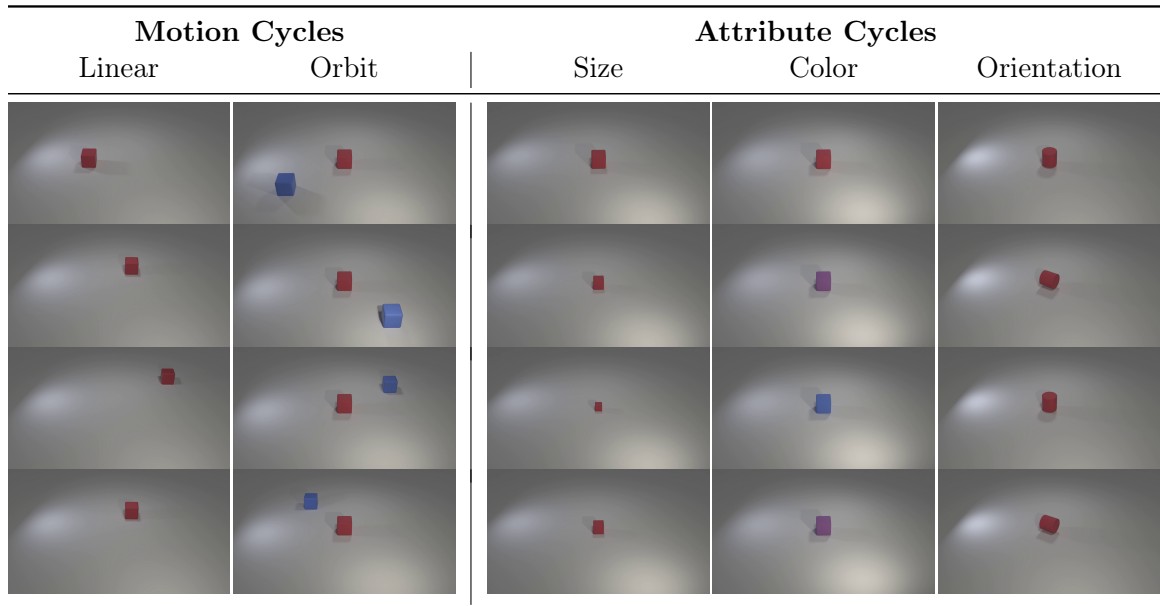

Table 1: **Time-dynamic objects via motion and attribute cycles:** CycliST integrates five types of object-centric state changes, each illustrated by a simple evolving scene with only that cycle applied. While motion cycles drive changes in the spatial relations between objects over time, attribute cycles introduce dynamics to the appearance of individual objects.

## 2 Cyclical State Transitions

We propose a scene-generation process for creating a benchmark dataset for VLMs revolving around Cyclical State Transitions (CycliST). CycliST is designed to challenge a model's understanding of cyclical patterns in high-resolution scenes using physically based rendering. We show CycliST's time-dynamic object descriptions (Sec. 2.1), the resulting motion (Sec. 2.2), attribute (Sec. 2.3), and light cycles (Sec. 2.4), as well as validation and rendering (Sec. 2.5).

### 2.1 Time-Dynamic Object Description

Each scene in the CycliST dataset contains a collection of $k$ objects, denoted by $\mathcal{O}$, where $k = i + j$. This collection consists of $i$ *cyclic objects* and $j$ *clutter objects*. While cyclic objects introduce structured temporal variation into the scene, clutter objects serve as fixed reference points that may still be referenced in questions.

Each object $o \in \mathcal{O}$ is defined as a tuple $(\mathcal{G}, s, m, c, \mathbf{p}, \boldsymbol{\omega}, \mathcal{C})$ comprising a mesh $\mathcal{G}$, a discrete size $s$ (either *small* or *large*), a material $m$ (either *metal* or *ruber*), a color $c$, a position $\mathbf{p}$, an orientation $\boldsymbol{\omega}$, and an associated set of cycle functions $\mathcal{C}$. For clutter objects, $\mathcal{C} = \emptyset$.

Cycle functions govern how a cyclic object evolves over time. Each function $f \in \mathcal{C}$ maps an initial object $o_0$, a time of $t$ seconds, and cycle frequency $\lambda$ in Hz, to a new configuration of that object at time $t$, such that $f$ is modifying one or more of its dynamic properties, namely, position, orientation, color, and size:

$$f(o_0, t, \lambda) = o_t.$$

In case of multiple functions acting upon the same object, we assert that none of them concurrently alter the same object property to avoid conflicts. Furthermore, we assure that

$$\forall l \in \mathbb{N} : f(o_0, l \cdot \lambda^{-1}, \lambda) = o_0,$$

i.e., after a full cycle, the object always returns to its initial state. We choose $\lambda$ based on a randomly chosen combination of prime factors of the scene's overall number of frames.

In CycliST, the object geometry and material remain fixed throughout time, leaving such transformations for future work. Although clutter objects do not exhibit time-dependent behavior themselves, they may still play a role in temporally grounded questions, since their spatial and visual relationships to dynamic objects evolve over time.

Objects in each scene are generated sequentially, one after the other. The discrete properties of each object—namely, its geometry and size—are sampled uniformly from the set of available options. Initial positions are drawn from a uniform distribution over the scene's $x$ and $y$ boundaries, with fixed $z = 0$, and initial orientations are sampled uniformly from the interval $[0°, 360°)$. As detailed later in Section 2.5, we ensure that the resulting scenes remain free of intersections.

We introduce temporal dynamics by randomly determining the number of clutter objects and cycle functions to be included in the scene. Clutter objects are then generated first, with no associated cycles, followed by the generation of cyclic objects. The latter are created by randomly assigning and combining cycle functions until all have been allocated.

As a result, while the number of clutter objects is predetermined, the number of cyclic objects is implicitly defined by the number and assignment of cycle functions. For example, if two distinct cycle functions are sampled, the resulting scene may include either one cyclic object with both functions or two cyclic objects, each associated with a single function.

We distinguish two types of object cycles: *motion cycles*, which affect an object's position $\mathbf{p}$, and *attribute cycles*, which alter its color $c$, size $s$, or orientation $\omega$. These two forms of temporal dynamics will be discussed in detail in the following two sections, respectively.

## 2.2 Motion Cycles

Each cyclic object in a scene may be assigned a *motion cycle*, which governs how its position changes over time. We consider two types of motion cycles in CycliST: **linear motion** and **orbiting motion** (Tab. 1 left). Both cycle types manipulate the position $\mathbf{p}$ of an object over time, thereby enabling time-dependent reasoning about spatial relations, such as answering questions about where two objects are relative to one another at a specific time step, or if one is ever or always in a specific relation to another.

We define **linear motion cycles** as a back-and-forth movement between the object's initial position $\mathbf{p}_0$ and a designated, randomized switch point $\mathbf{p}_s$ within the scene's boundaries. The object moves linearly from $\mathbf{p}_0$ to $\mathbf{p}_s$, reverses direction upon reaching $\mathbf{p}_s$, and returns to $\mathbf{p}_0$, completing one full cycle every $\frac{1}{\lambda}$ seconds. Hereby, the object's velocity is constant, such that it reaches the switch point $\mathbf{p}_s$ at the midpoint of the cycle.

**Orbiting motion cycles**, in contrast, introduce circular trajectories. First, a random clutter object or another cyclic object is selected as the *center object* about which the orbiting will occur. Second, a random orbit radius $r \in [r_{min}, r_{max}]$ and initial angle $\gamma \in [0, 360)$ are sampled, and the initial position of the orbiting object is updated to match the corresponding point on the orbital path, offset by the center object's initial position.

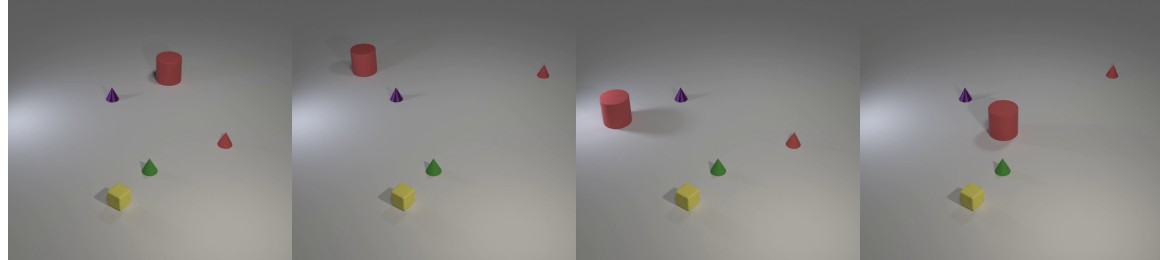

(a) A CycliST scene with a red cylinder orbiting a purple cone, and a red cone moving on a line.

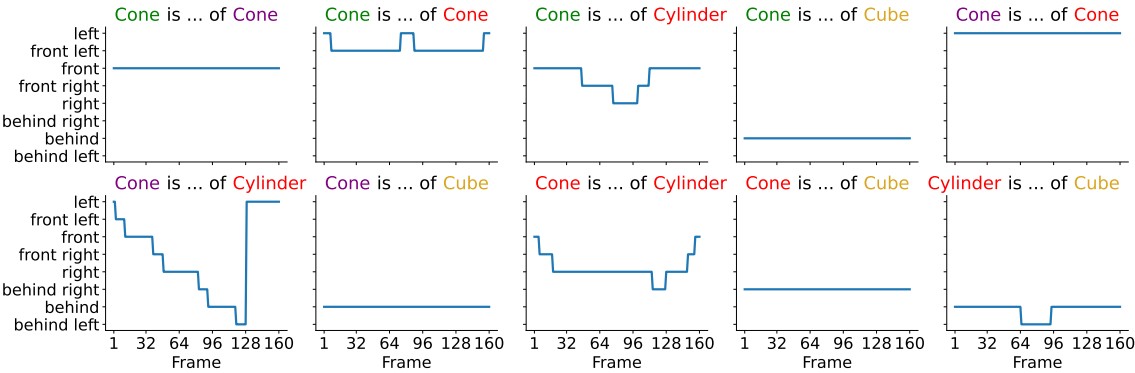

(b) Spatial relations of the scene in (a) over time.

Figure 2: **Visualizing cyclical state transitions in space:** We show the spatial relations between objects in an exemplary scene (a) over time (b). While some relations are constant, others are affected by the scene's motion cycles and exhibit a periodic pattern as well.

The position of an orbiting object is computed *relative to the center object's position at each time step*. This design enables stacked dynamics, where an object may orbit another that itself moves, e.g., in a linear motion cycle.

Figure 2 shows a CycliST scene with two motion cycles and illustrates how they affect the spatial relations between objects over time. Note that not only are the motion patterns cyclical, but the spatial relations exhibit an analogously periodic, although even more complex, trajectory. Furthermore, some relations within this dynamic scene remain unaffected by the embedded dynamics, allowing, for example, the posing of questions regarding a relation being *always* or *ever* true.

## 2.3 Attribute Cycles

In addition to spatial motion, **CycliST** incorporates *attribute cycles* that modify an object's visual characteristics over time (Tab. 1 right). These cycles affect properties such as an object's *size*, *orientation*, and *color*, thereby altering the object's appearance as the video progresses. As a result, an object's perceived identity evolves over time, increasing the complexity of scene interpretation and enabling a broader range of temporal reasoning questions. Hence, successfully answering such questions requires strong video understanding capabilities, including the ability to consistently track objects despite their visual changes.

The **size change cycle** function causes an object to alternate between two discrete size states, transitioning from *small* to *large* and back. To prevent visual artifacts such as ground penetration during resizing, the cycle function adjusts the object's vertical offset accordingly.

The **orientation change cycle** introduces continuous rotation by updating the object's placement angle over time. This cycle is applied only to objects whose rotation produces a visually significant change and is omitted for rotation-invariant geometries such as spheres.

Finally, the **color change cycle** performs continuous interpolation of the object's hue, resulting in smooth transitions between colors over time, e.g., cycling between red and blue.

Alongside spatial cycles, these attribute cycles provide CycliST with diverse temporal signals, significantly enriching scene dynamics and enabling the development of models capable of fine-grained temporal perception and object tracking.

## 2.4 Light Cycles

CycliST's lighting follows the CLEVR setup (Johnson et al., 2017). This includes randomizing light and camera positions with random translations to enhance visual diversity and reduce overfitting to fixed scene parameters. In addition to the cyclic motion and attribute changes we've introduced for individual objects, we also propose a scene-wide periodic lighting behavior. For each light source in the scene, we modulate its intensity using a sinusoidal pattern. This makes the lighting time-dependent, causing the scene to smoothly interpolate between bright and dark states. This synchronized, scene-wide change adds another layer of complexity on top of our motion and attribute cycles.

## 2.5 Scene Validation and Rendering

To ensure high-quality and physically plausible scenes, we employ a strict validation and rendering pipeline during scene generation. Each scene is validated incrementally at every generation step, allowing for early detection and correction of invalid configurations.

Object placement is performed sequentially, one object at a time, using a backtracking mechanism. As each object is introduced into the scene, spatial constraints are enforced to ensure sufficient margins to both the scene boundaries and all previously placed objects. This margin check helps avoid immediate overlaps or near-collisions.

Once all objects have been tentatively placed, the scene is simulated across all video frames using the specified object motion functions, e.g., orbiting, bouncing, or linear movement. During this simulation step, pairwise margin constraints are again evaluated for every object pair at each frame. If any collisions or spatial violations are detected due to a specific property, e.g., initial position or orbit radius, the corresponding property is resampled up to a predefined number of attempts.

If resampling a property does not resolve the conflict, the entire object is discarded and regenerated. This process itself is also bounded by a maximum number of attempts, after which the generation logic backtracks and attempts a different configuration for the prior object. If a generation exceeds a backtracking limit, it is considered failed for the given seed.

To ease the process of generating valid scenes without collisions, CycliST facilitates a larger overall scene compared to, e.g., CLEVR (Johnson et al., 2017), by pulling the camera's view away from the scene, creating more space for objects to move in without collisions or violating margins.

Figure 3: **CycliST's question categorization model:** We consider two question types: temporal descriptive and scene representative. The former challenges VLMs not only to understand a scene, but also to determine if an answer is always true or only at some point in time. The latter tasks VLMs with both understanding the presented cycles themselves and extracting quantitative properties, such as the number of cycles or their periodicity.

Once a valid scene configuration is finalized, the scene description is passed to Blender, utilizing the Cycles engine [‡] for photorealistic, physically based rendering. To this end, each object is equipped with the necessary material information, such as roughness and anisotropy values. We further define keyframes for each object's transformation and use Blender's built-in interpolation, reducing computation time while maintaining smooth motion.

The final output consists of a $1920 \times 1080$-pixel video sequence at 32 frames per second, along with a corresponding JSON file. We record the complete temporal metadata for the scene, including object attributes such as position, scale, rotation, and color over time. Additionally, we provide spatial relationships as a basis for generating VQA tasks.

## 3 Template-based Video Question Generation

We evaluate the ability of VLMs to recognize time-varying objects in an evolving scene, as well as their changing relations to one another. To this end, CyliST extends the template-based question-generation approach used in prior synthetic VQA work, such as CLEVR (Johnson et al., 2017) and CLEVERER (Yi et al., 2020). Specifically, we introduce a new set of temporal and cyclical operators for generating question-answer pairs as illustrated in Figure 3.

In template-based VQA, each question is formed by combining a template, which consists of question strings and placeholders, with a functional program, as exemplified in Figure 4. During generation, placeholders in the question strings, one for each attribute and cycle type (see Figure 4 left), such as $<C>$ for color or $$ for shape, are instantiated by sampling. For instance, the template "What is the size of the $<C>$ $$?" might become "What is the size of the blue cube?". A functional program is then executed on the scene's ground truth to compute the correct answer. To this end, we incorporate novel operators into the question-answer generation process for quantifying temporal questions. Namely, we introduce universal and existential quantifiers to the VQA pipeline to specify in questions and answers if a question is always true or at least once. Building on this foundation, CycliST incorporates novel question strings and placeholders that facilitate a wider range of reasoning about temporal properties (see Section 3.1) and scene properties (see Section 3.2).

---

[‡] https://www.cycles-renderer.org/

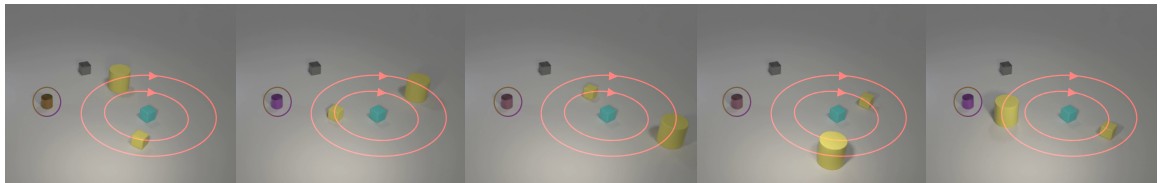

(a) A scene with two orbits and one color change cycle.

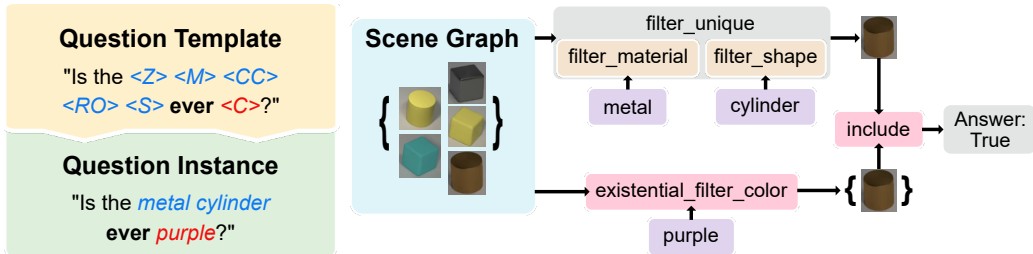

(b) An existential, temporal descriptive question (query).

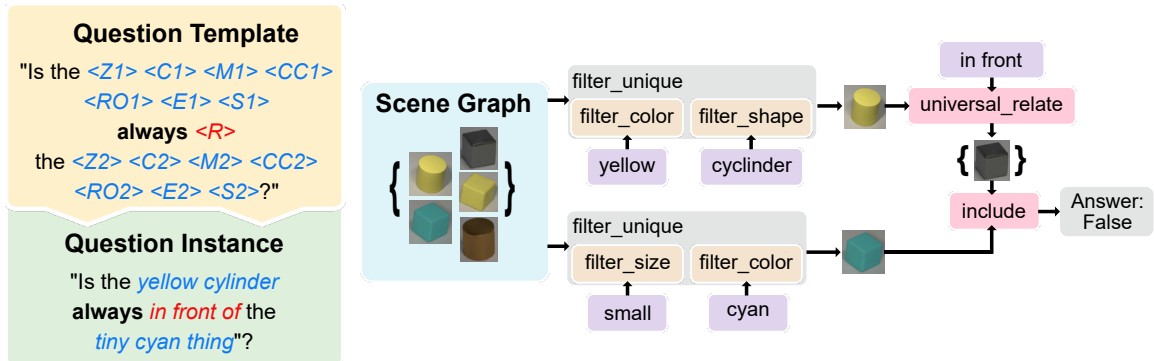

(c) A universal, temporal descriptive question (relate).

Figure 4: **CycliST's question generation pipeline:** Given a question template and scene graph, CycliST samples a question instance and yields a ground-truth answer through a functional program applied to the scene graph. Here, two example question-answer pairs are derived based on a CycliST scene (a), one being an existential temporal query (b) and one universal temporal relation (c).

We exemplify CycliST's VQA pipeline on a scene and two question templates in Figure 4. Furthermore, the full set of question strings and placeholders employed in CycliST is provided in Table 13 in the Appendix.

## 3.1 Temporal Descriptive VQA

With temporal descriptive VQA, CycliST enables the evaluation of a model's temporal understanding by probing object properties and relationships over time. To achieve this, we employ two temporal quantifiers: **universal** ($\forall$), which asks about properties that are *always* true, and **existential** ($\exists$), which asks if properties are *ever* true.

Answering a universal question requires tracking attributes throughout the entire video, whereas an existential question only requires finding a single corresponding frame. These quantifiers are applied across three distinct subcategories of questions:

**Query** These questions probe whether a single object exhibits a specific attribute over time. Attributes can be either static (e.g., material, shape) or dynamic (e.g., color, size, position) depending on the applied cycles. For an object with dynamic attributes, a universal question ($\forall$) requires the attribute to persist throughout the video, while an existential question ($\exists$) requires it to appear at least once.

**Compare** These questions involve comparing an attribute between two objects. A universal comparison requires the relationship to hold continuously. For instance, answering "Is the blue cube always the same size as the red sphere?" requires both objects to be either static and the same size or have perfectly synchronized size-change cycles. An existential comparison, however, only requires the condition to be met at least once.

**Relate** These questions focus on the relative spatial relationship between two objects. Again, a universal question demands that a given relationship (e.g., "left of") holds throughout the video, while an existential question requires it to hold in at least one frame.

Generating appropriate temporal descriptive question-answer pairs requires employing the appropriate quantifiers during the generation process. For example, CycliST may select all objects that are "*ever* blue" ($\exists$) or all objects that are "*always* small" ($\forall$).

## 3.2 Scene Representative VQA

In contrast to temporal questions that evaluate properties over time, Scene Representative VQA assesses a model's ability to understand the scene's overall composition and cyclic nature. This category is divided into two main types of questions: cyclic and numeric.

**Cyclic** Cyclic questions probe the specific parameters and nature of the cyclical transformations applied to objects. These can be further broken down into several types. **Orbit** questions focus on the motion cycles of the same name, asking, for example, whether an object is being orbited by another or to identify the center of an object's orbit. **Initial** questions query the visual attribute of an object at the beginning of the video, such as its initial size or color. Finally, **transition** questions test the understanding of attribute changes by asking what size or color an object is transitioning into.

**Numeric** Numeric questions require the model to quantify aspects of the scene and its dynamics. This includes **counting** questions (e.g., "How many cyclic objects are there?"), questions about **periodicity** (e.g., "What is the period of an object's color change cycle?"), and questions about **occurrence**, which require counting how many times a specific event happens.

In tandem with temporal descriptive questions, scene representative questions ensure that the VLMs can obtain a complete and correct representation of the generated set of objects and dynamics, and that correct conclusions about the emergent behavior can be drawn. Note that not all questions will refer to all cycles, e.g., cycle transition questions do not ask about orientation change cycles as they do not exhibit the necessary visual reference points.

| | Time | | | |
|---|---|---|---|---|
| **Tier** | 0 s | 1.25 s | 2.5 s | 3.75 s |

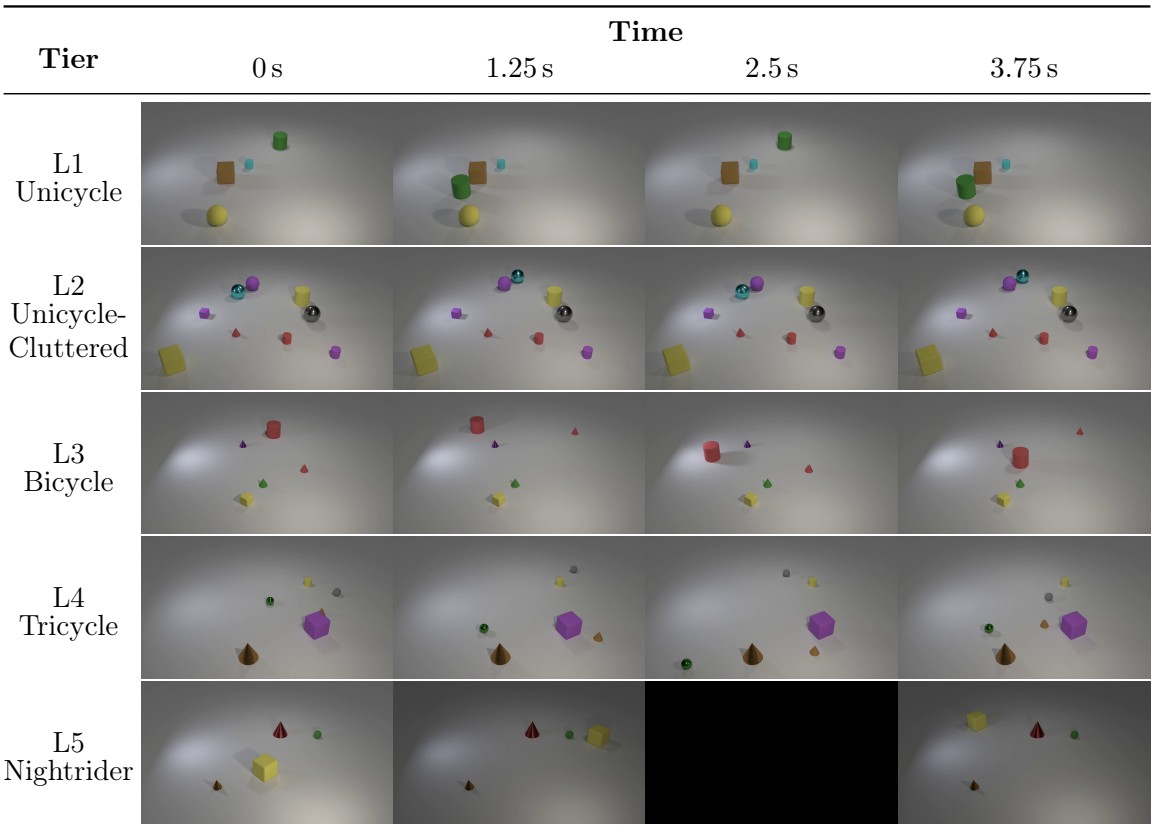

Table 2: **Tiered benchmarking in CycliST:** CycliST provides multiple levels of difficulty for Video Question Answering and Scene Understanding. Here, each row shows an example scene from each benchmark tier evolving over time.

## 4 Tiered Scene Construction and Benchmarking

To provide a rigorous and informative evaluation of VLMs on CycliST scenes, we propose a benchmark comprising multiple difficulty tiers as seen in Table 2, rather than relying on a single setting across all of CycliST's samples. Each tier tests VLMs on their ability to handle increasing levels of visual complexity, temporal reasoning, and contextual inference.

The key differentiating factors across tiers include the number of cyclic objects in a scene, the amount of visual clutter introduced by static objects, and variations in scene lighting conditions. These changes are designed to test specific capabilities of VQA systems, such as their ability to track motion patterns, isolate relevant entities amidst noise, and answer complex questions.

**L1 - Unicycle** This introductory tier assesses a VLM's fundamental ability to detect and describe periodic motion or attribute changes. Each scene contains exactly one cyclic object that undergoes a variable number of full cycles, accompanied by 2 to 3 clutter objects. This setup establishes a minimal yet controlled environment to verify that a model can perceive and reason over temporal patterns and the different types of cycle functions in isolation.

Table 3: **CycliST comprises 14.8k FHD videos.**

| Tier | Training | Testing | Validation | Total |
|---|---|---|---|---|
| L1 - Unicycle | 1500 | 750 | 750 | 3000 |
| L2 - Unicycle-Cluttered | 1500 | 750 | 750 | 3000 |
| L3 - Bicycle | 1500 | 750 | 750 | 3000 |
| L4 - Tricycle | 1540 | 770 | 770 | 3080 |
| L5 - Nightrider | 1360 | 680 | 680 | 2720 |
| **Total** | 7400 | 3700 | 3700 | **14,800** |

**L2 - Unicycle-Cluttered** This tier introduces additional background complexity by increasing the number of clutter objects to 4-9 per scene. Although only a single cyclic object remains, the added visual complexity significantly raises the difficulty and expands the space of possible questions. This setting assesses a model's robustness in answering questions about specific objects within a larger group of candidates.

**L3 - Bicycle & L4 - Tricycle** These two splits further escalate the reasoning challenge by including multiple cyclic objects: two in the Bicycle split and three in the Tricycle split. In both benchmark tiers, two to three clutter objects are present in each scene. To answer questions in L3 and L4, a model must not only understand scenes with many objects but also demonstrate robustness to multiple cycles, which significantly increases scene complexity. This entails understanding how the interactions and relative phases of these cycles relate to one another over time, which requires a more sophisticated understanding of spatiotemporal dependencies.

**L5 - Nightrider** Our final benchmark tier uses the Light Cycle introduced in Section 2.4, making it more challenging to track objects and their attributes throughout the video. To this end L5, comprises a balanced variation of scenes from L1, L3, and L4.

Each CycliST tier changes one scene aspect, namely the amount of visual clutter (L2), the number of cycles per video (L3, L4), and the introduction of lighting cycles. We provide a balanced number of samples divided into training, validation, and test splitsas detailed in Table 3. Furthermore, for each scene, we aim to generate one question per template, totaling about 120k questions (see Table 14 in the Appendix for a detailed overview of test split questions). Note that not all templates apply to all scenes; for example, one cannot ask about orbits in a scene with linear motion only.

## 5  Video Language Models on CycliST

Our experimental section aims to evaluate the performance of state-of-the-art VLMs on cyclic video data. To this end, we consider the VQA and scene understanding capabilities of VLMs ranging from smaller 7B to larger 78B models. We cover both proprietary and open-source models as outlined in Section 6, specifically from the Intern (Wang et al., 2025; Zhu et al., 2025), LLaVA-Video (Zhang et al., 2025), LLaVA-OV (Li et al., 2025), and Gemini (Comanici et al., 2025) families.

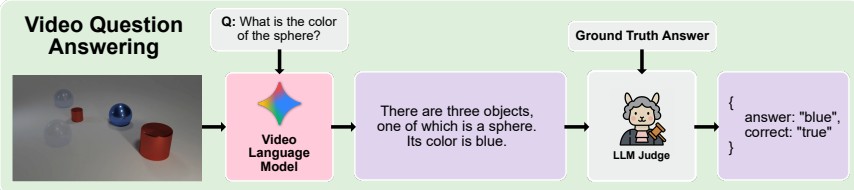

(a) LLM-judge driven VQA pipeline.

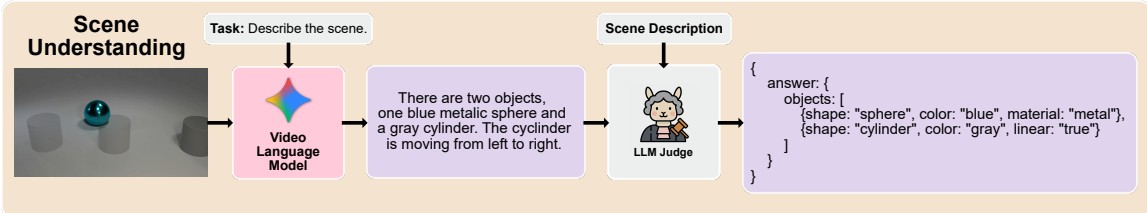

(b) LLM-judge driven scene understanding pipeline.

Figure 5: **CycliST's evaluation pipelines:** By employing an LLM-judge in its VQA (a) and scene understanding (b) pipelines, VLMs can give free-form answers rather than being limited to a multiple-choice questionnaire.

The models are provided with the complete 5-second video, which consists of 160 frames. For the LLaVA-Video model family, we provide half the frames, since they are limited in the context size they can handle. Nevertheless, they maintain competitive performance.

Since VLMs provide free text answers, we employ Llama3-70B as a judge (Zheng et al., 2023) as illustrated in Figure 5 for its robustness to phrasing variations compared to methods such as BLEU-4 (Papineni et al., 2002) and ROUGE (Lin, 2004). To ensure correct judgments, we employ 100 calibration questions for each judgment pipeline. More specifically, we treat four different cases: The judge may (i) extract "Yes" or "No", (ii) numeric values, (iii) attributes, or (iv) map recognized objects into a pre-defined JSON format. In the first two cases, we achieve 100% alignment between LLM and human judgments. In the third case, we obtained a lower alignment of 92.6% because the judge had to address ambiguities arising from synonymous descriptions of the same attributes. Finally, we obtain an F1 score of 87.6% on object mapping for scene understanding, since in this case the judge may report more or fewer objects than are present in the human annotation. We provide the scripts used to calibrate Llama3's judgment alongside the generation code.

When the VLM produces an indefinite answer, we tag it in the JSON and consider it incorrect when computing accuracy. In metrics such as the Mean Absolute Error (MAE), these answers are excluded. Where applicable, we highlight random guessing as baseline performance as the lower end of the color bands in the tables.

## 5.1 Temporal Descriptive VQA

We assess the temporal reasoning capabilities of the selected VLMs on a set of yes/no questions. First, query questions test the ability to extract scene attributes. Second, comparison (comp) questions require matching the attributes of two objects. Third, relational (rel) questions assess if a spatial relationship holds between objects.

Table 4: **VLMs struggle at Temporal Descriptive VQA:** Oftentimes, the accuracy of the tested models is only slightly above random guessing (50%), with peak performance not exceeding 83%. We show results on existential ($\exists$) and universal ($\forall$) temporal questions.

| | | Temporal Descriptive ( 50% ▬▬ 100%) | | | | | | | | |
|---|---|---|---|---|---|---|---|---|---|---|
| | | Unicycle | | | Bicycle | | | Tricycle | | |
| Q | Model | query | comp | rel | query | comp | rel | query | comp | rel |
| | InternVideo2.5 (8B) | 72.9 | 67.4 | 74.8 | 73.2 | 63.0 | 72.8 | 70.1 | 63.7 | 70.4 |
| | InternVL3 (8B) | 79.2 | 72.6 | 74.0 | 76.8 | 68.8 | 73.0 | 75.5 | 66.3 | 72.3 |
| | InternVL3 (78B) | 73.5 | **74.7** | **80.0** | 73.5 | **74.1** | **79.3** | 73.1 | **72.4** | **77.1** |
| | LLaVA-Video (7B) | 60.7 | 56.2 | 58.2 | 61.7 | 54.7 | 58.4 | 60.7 | 53.7 | 53.8 |
| $\exists$ | LLaVA-Video (72B) | **80.4** | 70.0 | 70.1 | **79.1** | 67.8 | 69.6 | **78.4** | 66.5 | 65.7 |
| | LLaVA-OV (7B) | 73.5 | 58.9 | 62.0 | 71.1 | 57.1 | 60.8 | 71.3 | 56.0 | 61.1 |
| | LLaVA-OV (72B) | 74.8 | 63.1 | 62.8 | 73.3 | 62.8 | 64.5 | 72.8 | 60.4 | 62.9 |
| | Gemini Flash 2.0 | 75.7 | 64.5 | 61.3 | 72.1 | 62.2 | 65.2 | 73.1 | 62.5 | 60.1 |
| | Gemini Flash 2.5 | 73.0 | 65.9 | 73.4 | 73.6 | 67.0 | 74.0 | 71.9 | 64.5 | 73.8 |
| | InternVideo2.5 (8B) | **83.2** | 63.7 | 77.0 | 81.7 | 64.3 | 73.9 | 81.9 | 63.8 | 75.9 |
| | InternVL3 (8B) | 77.0 | 69.9 | 73.9 | 77.0 | 67.5 | 74.7 | 77.0 | 65.5 | 75.8 |
| | InternVL3 (78B) | 75.1 | 72.5 | **79.8** | 74.9 | **71.7** | **79.1** | 77.0 | **69.6** | **80.0** |
| | LLaVA-Video (7B) | 66.8 | 61.4 | 57.2 | 65.6 | 56.6 | 55.3 | 61.7 | 55.6 | 58.0 |
| $\forall$ | LLaVA-Video (72B) | 82.5 | 71.3 | 75.4 | 79.9 | 67.8 | 72.5 | 82.4 | 67.3 | 72.1 |
| | LLaVA-OV (7B) | 76.0 | 64.0 | 62.9 | 74.7 | 62.4 | 60.3 | 76.6 | 58.9 | 63.1 |
| | LLaVA-OV (72B) | 79.3 | **73.7** | 72.6 | 78.4 | 66.4 | 69.1 | 79.2 | 66.9 | 69.1 |
| | Gemini Flash 2.0 | 63.5 | 67.2 | 66.5 | 66.3 | 65.3 | 63.7 | 68.1 | 62.8 | 61.9 |
| | Gemini Flash 2.5 | 77.6 | 72.3 | 79.6 | **81.8** | 69.5 | 75.7 | **82.9** | 68.2 | 77.6 |

**Q1: Can VLMs detect objects and their relations in a time-dynamic scene?** We present the accuracies for all models and both quantifiers in Table 4, with the baseline being 50% due to a balanced answer distribution (yes/no). Overall, query performance proved most consistent across all VLMs. This follows from queries being similar to past benchmarks, on which state-of-the-art models may already have been trained.

Interestingly, some models perform above average in specific categories, such as LLaVA-Video (72B) on existential-attribute questions, but not on universal questions or other question categories. The best-performing model (InternVL3 (78B) ) is outperforming the others in 11/18 categories, with most declining in performance as tier difficulty increases.

While the best models achieve only 70–80% accuracy on query and relational questions, they exhibit a much larger technical gap on comparison questions. Hence, effective temporal and inter-object reasoning remains a significant challenge for VLMs.

**Q2: How does added clutter and light-cycles affect VLMs' temporal descriptive capabilities?** We compare the accuracy on Unicyle with its cluttered variant and the overall performance on Nightrider. One can see how added clutter is causing a performance hit to almost all models on all question types (Table 5 left). Although there is not the same performance drop when looking at the Nightrider accuracies (Table 5 right), the tested VLMs struggle to solve this tier similarly to Table 4.

Table 5: **Applying clutter and light-cycles to the scenes**: Accuracy changes in percentage points are shown on the left, results for Nightrider are displayed on the right (baseline at 50%). We show results on existential (∃) and universal (∀) temporal questions.

| Q | Model | Uni. clut. (-10 to +10) query | comp | rel | Q | Model | N.R. ( 50% to 100%) query | comp | rel |
|---|---|---|---|---|---|---|---|---|---|
| ∃ | InternVideo2.5 (8B) | -2.8 | -3.4 | +1.5 | ∃ | InternVideo2.5 (8B) | 71.4 | 67.9 | 73.8 |
|  | InternVL3 (8B) | -4.2 | -4.2 | +1.1 |  | InternVL3 (8B) | 74.3 | 66.7 | 70.8 |
|  | InternVL3 (78B) | +2.8 | **+0.0** | **+3.5** |  | InternVL3 (78B) | 72.5 | **72.8** | **79.3** |
|  | LLaVA-Video (7B) | -4.9 | -2.0 | -1.5 |  | LLaVA-Video (7B) | 63.5 | 53.5 | 56.3 |
|  | LLaVA-Video (72B) | -2.8 | -7.6 | -6.2 |  | LLaVA-Video (72B) | **79.9** | 70.3 | 70.0 |
|  | LLaVA-OV (7B) | -5.8 | -1.6 | -1.9 |  | LLaVA-OV (7B) | 73.6 | 56.8 | 60.9 |
|  | LLaVA-OV (72B) | -6.9 | -6.1 | -2.8 |  | LLaVA-OV (72B) | 74.4 | 63.7 | 66.8 |
|  | Gemini Flash 2.0 | -4.5 | -2.7 | +1.2 |  | Gemini Flash 2.0 | 72.8 | 60.2 | 60.4 |
|  | Gemini Flash 2.5 | **+3.3** | -1.9 | -0.8 |  | Gemini Flash 2.5 | 71.4 | 64.5 | 70.6 |
| ∀ | InternVideo2.5 (8B) | -3.6 | -5.3 | -7.4 | ∀ | InternVideo2.5 (8B) | **81.6** | 65.1 | 75.8 |
|  | InternVL3 (8B) | +0.3 | -3.7 | **+0.8** |  | InternVL3 (8B) | 73.1 | 73.1 | 74.3 |
|  | InternVL3 (78B) | +1.3 | **-0.8** | -1.5 |  | InternVL3 (78B) | 73.3 | 70.0 | **80.6** |
|  | LLaVA-Video (7B) | -5.9 | -6.7 | -2.7 |  | LLaVA-Video (7B) | 63.7 | 54.3 | 55.9 |
|  | LLaVA-Video (72B) | -4.4 | -9.9 | -8.1 |  | LLaVA-Video (72B) | 80.8 | **73.3** | 74.6 |
|  | LLaVA-OV (7B) | -3.6 | -4.8 | -2.5 |  | LLaVA-OV (7B) | 74.6 | 65.7 | 61.7 |
|  | LLaVA-OV (72B) | -2.9 | -8.9 | -4.9 |  | LLaVA-OV (72B) | 76.1 | 70.8 | 70.6 |
|  | Gemini Flash 2.0 | +0.8 | -3.9 | -8.0 |  | Gemini Flash 2.0 | 63.9 | 60.7 | 58.8 |
|  | Gemini Flash 2.5 | **+2.4** | -4.9 | -4.9 |  | Gemini Flash 2.5 | 78.4 | 68.9 | 74.5 |

Table 6: **VLMs fail to understand orbits**: Models guess orbit directions and center object attributes with low accuracy (random baseline about 50%/30% accuracy, respectively).

| Model | Direction (50% to 100%) Uni. | Bi. | Tri. | Uni. clut. | N.R. | Center (30% to 100%) Uni. | Bi. | Tri. | Uni. clut. | N.R. |
|---|---|---|---|---|---|---|---|---|---|---|
| InternVideo2.5 (8B) | 46.7 | 42.7 | 44.8 | 49.2 | 38.8 | 50.7 | 33.3 | 31.4 | 29.9 | 36.5 |
| InternVL3 (8B) | 52.6 | **51.6** | 50.0 | 50.0 | **49.6** | **60.1** | **51.4** | **48.9** | **54.9** | **50.8** |
| InternVL3 (78B) | 48.2 | 43.0 | 44.8 | 49.2 | 46.7 | 57.8 | 43.2 | 42.3 | 40.5 | 45.4 |
| LLaVA-Video (7B) | 40.6 | 37.3 | 38.0 | 36.7 | 37.1 | 39.1 | 34.7 | 31.5 | 29.1 | 35.0 |
| LLaVA-Video (72B) | 51.8 | 49.3 | 51.6 | 51.0 | 45.8 | 55.4 | 50.6 | 47.1 | 40.8 | 46.9 |
| LLaVA-OV (7B) | 53.3 | 48.8 | 49.0 | 46.1 | 49.2 | 46.0 | 41.4 | 45.5 | 25.8 | 40.4 |
| LLaVA-OV (72B) | **54.1** | 49.3 | **52.6** | **53.9** | 49.2 | 48.0 | 41.0 | 42.2 | 33.8 | 48.5 |
| Gemini Flash 2.0 | 42.2 | 41.8 | 46.4 | 46.9 | 34.6 | 25.0 | 22.3 | 19.4 | 16.8 | 23.2 |
| Gemini Flash 2.5 | 49.6 | 44.3 | 48.7 | 47.5 | 49.4 | 46.5 | 34.5 | 35.5 | 40.0 | 34.4 |

## 5.2 Scene Representative VQA

**Q3: Can VLMs detect the center or direction of an orbit?** When questioned about the direction of an orbit, possible answers are clockwise or counterclockwise. Models are randomly guessing at or below 50% accuracy as shown in Table 6. If a model does not provide a direction as an answer, we score this as a wrong answer. This is especially evident with smaller models, such as LLaVA-Video (7B) , which exhibit accuracies significantly below random guessing. When asked about the properties of the center object of the orbits, we achieve accuracies of up to 60.1% on Unicycle.

Table 7: **VLMs show moderate performance when queried about changing attributes**: Performance does not exceed 66.3% accuracy, with the best results on Unicycle.

| Model | Cyclic Transition (30% ▨ 100%) | | | | |
|---|---|---|---|---|---|
| | Unicycle | Bicycle | Tricycle | Uni. clut. | Nightrider |
| InternVideo2.5 (8B) | 45.2 | 40.5 | 36.9 | 35.3 | 35.4 |
| InternVL3 (8B) | **66.3** | **63.7** | **59.9** | **62.9** | **60.2** |
| InternVL3 (78B) | 59.4 | 59.6 | 54.7 | 58.0 | 58.3 |
| LLaVA-Video (7B) | 55.1 | 54.8 | 46.2 | 44.4 | 50.0 |
| LLaVA-Video (72B) | 60.6 | 58.8 | 53.5 | 44.5 | 52.7 |
| LLaVA-OV (7B) | 52.3 | 50.1 | 47.3 | 39.9 | 48.3 |
| LLaVA-OV (72B) | 50.5 | 52.5 | 44.3 | 41.0 | 48.5 |
| Gemini Flash 2.0 | 49.5 | 51.8 | 42.6 | 41.3 | 44.9 |
| Gemini Flash 2.5 | 45.0 | 50.8 | 48.2 | 47.3 | 42.8 |

Table 8: **VLMs struggle with counting object cycles**: Accuracy and MAE by model on cycle counting. An MAE of one means miscounting by one cycle on average.

| Model | Accuracy (0% ▨ 100%) | | | MAE (>1 ▨ 0) | | |
|---|---|---|---|---|---|---|
| | Unicycle | Bicycle | Tricycle | Unicycle | Bicycle | Tricycle |
| InternVideo2.5 (8B) | 50.2 | 34.8 | 31.1 | 0.57 | 0.80 | 0.96 |
| InternVL3 (8B) | 55.4 | 41.7 | 34.6 | 0.45 | 0.72 | 0.92 |
| InternVL3 (78B) | 55.4 | 46.5 | 37.8 | 0.66 | 0.81 | 0.98 |
| LLaVA-Video (7B) | 15.4 | 19.9 | 19.8 | 1.70 | 1.60 | 1.80 |
| LLaVA-Video (72B) | 15.3 | 15.2 | 12.0 | 2.00 | 2.18 | 2.59 |
| LLaVA-OV (7B) | 8.6 | 21.2 | 25.2 | 1.61 | 1.31 | 1.31 |
| LLaVA-OV (72B) | 26.0 | 19.1 | 18.3 | 1.58 | 1.77 | 1.98 |
| Gemini Flash 2.0 | 55.1 | 42.4 | 33.6 | 0.52 | 0.75 | 1.01 |
| Gemini Flash 2.5 | **69.4** | **60.5** | **46.1** | **0.32** | **0.48** | **0.75** |

Note that in this category, not all questions have only two possible answers, since there are 8 color options and 2 size options. Instead, random guessing yields approximately 30% accuracy. Due to an increasing number of orbits in Bicycle and Tricycle scenes, respectively, we see a performance drop for most models. Similar to **Q2**, we see a stark performance drop when clutter is added, and similar performance on Nightrider since it does not increase the number of orbits and contains simpler, Uni- and Bicycle-like scenes. Overall, these results suggest that current VLMs can not infer orbit directions and struggle to identify their centers.

**Q4: Can VLMs track attribute changes?** In this category, models are asked to identify objects that change their color or size and are queried about the initial attribute value or the value they transition into. Again, as in **Q3**, random guessing would yield approximately 30% accuracy. Table 7 shows the accuracy of detecting the correct value. The effect of added clutter on Unicycle, as well as the performance on Nightrider, is similar to Table 6 right. To summarize, the models' visual recognition of dynamic object sizes and colors is lacking.

Table 9: **VLMs fail at counting completed cycles.** All model predictions are, on average, off by more than 1, with possible answers being one, two, or five completed cycles (depending on the randomly chosen number of prime factors of the overall number of frames).

| Model | Accuracy (0% ▓▓ 100%) | | | MAE (>1 ▓▓ 0) | | |
| --- | --- | --- | --- | --- | --- | --- |
| | Unicycle | Bicycle | Tricycle | Unicycle | Bicycle | Tricycle |
| InternVideo2.5 (8B) | **46.4** | **38.7** | 32.9 | **0.92** | **1.28** | **1.61** |
| InternVL3 (8B) | 34.7 | 27.9 | 30.0 | 1.17 | 1.59 | 1.71 |
| InternVL3 (78B) | 31.4 | 27.3 | 27.6 | 1.34 | 1.60 | 1.78 |
| LLaVA-Video (7B) | 42.7 | 31.6 | **34.0** | 1.54 | 1.65 | 1.77 |
| LLaVA-Video (72B) | 32.2 | 23.7 | 30.7 | 1.45 | 1.92 | 1.57 |
| LLaVA-OV (7B) | 42.3 | 33.1 | 32.6 | 1.31 | 1.51 | **1.61** |
| LLaVA-OV (72B) | 21.1 | 16.9 | 18.3 | 3.31 | 3.36 | 3.28 |
| Gemini Flash 2.0 | 15.2 | 22.8 | 19.2 | 1.70 | 1.84 | 2.01 |
| Gemini Flash 2.5 | 30.1 | 20.9 | 16.3 | 1.44 | 2.05 | 2.27 |

Table 10: **VLMs have no notion of frames per second:** Objects can complete their cycles within 160, 80, or 32 frames. When asking the VLMs about the number of frames it took for a specific cycle to complete, all predictions are off by a large margin.

| Model | MAE (>100 ▓▓ 0) | | |
| --- | --- | --- | --- |
| | Unicycle | Bicycle | Tricycle |
| InternVideo2.5 (8B) | 105.0 | 97.3 | 95.9 |
| InternVL3 (8B) | 102.8 | 96.2 | 90.2 |
| InternVL3 (78B) | **90.5** | **88.1** | **87.2** |
| LLaVA-Video (7B) | 105.1 | 98.9 | 95.4 |
| LLaVA-Video (72B) | 97.8 | 107.7 | 92.3 |
| LLaVA-OV (7B) | 101.7 | 93.6 | 92.5 |
| LLaVA-OV (72B) | 103.2 | 95.4 | 93.9 |
| Gemini Flash 2.0 | 105.4 | 98.7 | 96.7 |
| Gemini Flash 2.5 | 109.9 | 89.6 | 89.7 |

**Q5: Can VLMs count the occurrence of cyclic transitions?** Lastly, we evaluate whether VLMs can count the number of cycles in a scene. In Table 8 we report two metrics for this template: accuracy, which measures how often a model predicts the exact number of cycles, and the average absolute error, i.e., the distance between the predictions and the ground truth. The performance varies strongly among the models, with Gemini Flash 2.5 taking the lead. Possible answers range from zero to one for a Unicycle, to two for a Bicycle, and to three for a Tricycle scene. On Unicycle, Gemini Flash 2.5 is the only model that clearly outperforms random guessing, while all other models even perform below random guessing. A possible explanation is that models predict a number based on a learned bias, as they are unable to count cycles in the video. The same observations hold for bicycles and tricycles, as seen in the decreasing accuracy with increasing numbers of objects to count.

Table 11: **VLMs perform well on scene captioning in CycliST:** Precision, recall, and F1 scores are reported for matching objects by their attributes with the respective ground truth scene description.

| | Scene Captioning (0% ▬▬ 100%) | | | | | | | | |
| | Unicycle | | | Bicycle | | | Tricycle | | |
| Model | Pr | Re | F1 | Pr | Re | F1 | Pr | Re | F1 |
|---|---|---|---|---|---|---|---|---|---|
| InternVideo2.5 (8B) | 83.0 | 78.9 | 80.4 | 82.1 | 76.0 | 78.4 | 82.8 | 77.1 | 79.4 |
| InternVL3 (8B) | **92.8** | 88.4 | 89.9 | 89.5 | 84.4 | 86.1 | 89.1 | 82.6 | 84.8 |
| InternVL3 (78B) | 92.1 | **93.1** | **92.5** | **89.9** | **91.5** | **90.4** | **90.3** | **91.7** | **90.6** |
| LLaVA-Video (7B) | 79.5 | 76.5 | 77.7 | 78.0 | 73.5 | 75.2 | 77.9 | 72.3 | 74.5 |
| LLaVA-Video (72B) | 86.0 | 85.7 | 85.7 | 84.0 | 83.0 | 83.2 | 82.5 | 80.0 | 80.8 |
| LLaVA-OV (7B) | 78.8 | 79.0 | 78.7 | 79.8 | 79.1 | 79.1 | 79.6 | 78.5 | 78.7 |
| LLaVA-OV (72B) | 82.9 | 81.9 | 82.3 | 81.0 | 79.7 | 80.1 | 80.3 | 79.6 | 79.7 |
| Gemini Flash 2.0 | 81.6 | 82.7 | 82.0 | 79.8 | 81.5 | 80.4 | 78.9 | 81.1 | 79.8 |
| Gemini Flash 2.5 | 75.4 | 76.8 | 75.9 | 74.0 | 77.2 | 75.2 | 72.7 | 76.4 | 74.2 |

Table 12: **VLMs fail to caption cyclic transitions:** Precision, recall, and F1 scores are reported for matching cyclic transitions of matched and unmatched objects with the respective ground truth scene description.

| | Cycle Captioning (0% ▬▬ 100%) | | | | | | | | |
| | Unicycle | | | Bicycle | | | Tricycle | | |
| Model | Pr | Re | F1 | Pr | Re | F1 | Pr | Re | F1 |
|---|---|---|---|---|---|---|---|---|---|
| InternVideo2.5 (8B) | 13.9 | 17.2 | 14.9 | 19.7 | 14.5 | 15.8 | 23.5 | 14.5 | 16.7 |
| InternVL3 (8B) | 18.9 | 21.7 | 19.7 | 23.3 | 19.9 | 20.6 | 27.2 | 19.3 | 21.6 |
| InternVL3 (78B) | 24.5 | 26.2 | 25.1 | 33.3 | 23.0 | 26.1 | 35.1 | 18.0 | 22.8 |
| LLaVA-Video (7B) | 5.9 | 8.1 | 6.5 | 9.8 | 8.5 | 8.3 | 11.1 | 7.5 | 8.0 |
| LLaVA-Video (72B) | 10.7 | 13.0 | 11.3 | 14.6 | 11.3 | 11.9 | 15.4 | 10.1 | 11.1 |
| LLaVA-OV (7B) | 3.0 | 5.2 | 3.6 | 4.7 | 4.3 | 4.3 | 5.5 | 4.3 | 4.5 |
| LLaVA-OV (72B) | 1.6 | 1.9 | 1.7 | 2.9 | 2.5 | 2.4 | 4.1 | 2.8 | 3.0 |
| Gemini Flash 2.0 | 13.2 | 14.9 | 13.7 | 23.1 | 16.4 | 18.1 | 26.3 | 14.6 | 17.4 |
| Gemini Flash 2.5 | **28.3** | **32.2** | **29.4** | **36.8** | **30.2** | **31.2** | **38.8** | **27.7** | **30.0** |

**Q6: Can VLMs count completed cycles?**  Rather than counting specific cycles in the video, we evaluate the model's capabilities to detect and count cycles within the video, for example, how many times an object orbits around its center. Table 9 shows accuracies and MAE, and how all tested models fail at counting the cycle passes (MAEs at or above one).

**Q7: Can VLMs count the number of frames a cycle takes to complete?**  As shown in Table 10, all models have an accuracy of less than 0.5% and an MAE of approximately 100 frames. This is due to the fact that, upon introspecting the individual answers, we can see that all models in almost all questions predict a number of frames within the range of 0 to 20. This explains the average MAE of around 100, which is roughly the difference between the average ground truth and the prediction interval. Hence, the tested VLMs are unable to sum up the number of frames within which an event occurs, and may be biased by the frame number observed during training.

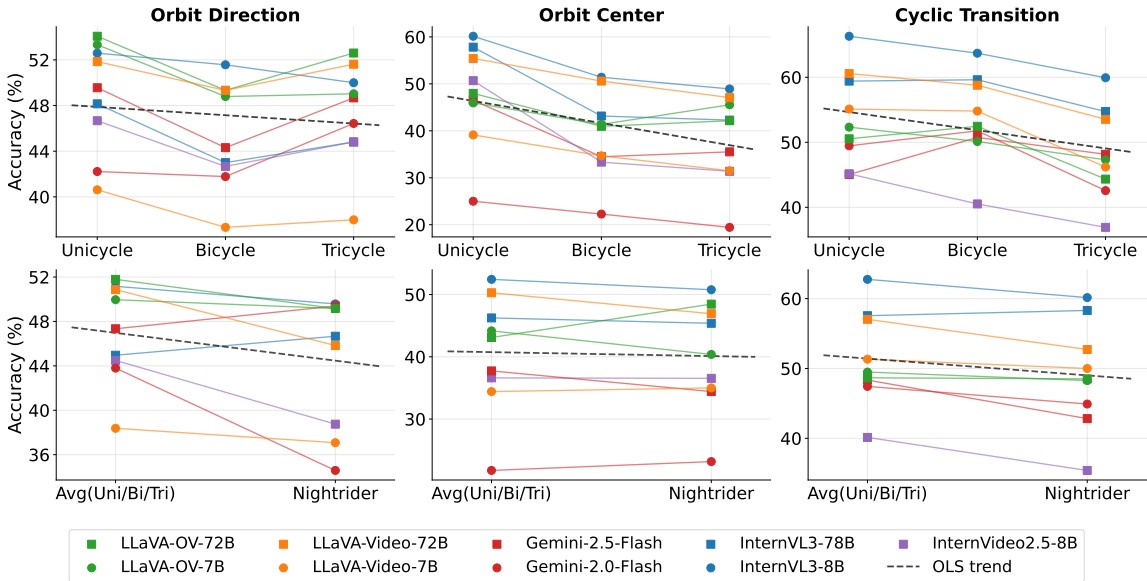

Figure 6: **Decomposition of performance drop across tiers:** We show that, when controlling for clutter, scene-representative performance deteriorates with the addition of object cycles (upper row) or light cycles (lower row), respectively. Color indicates model family, marker shape indicates model version. The dashed line indicates the average decline in performance via Ordinary Least Squares (OLS) regression.

**Q8: How do cycle count and light cycles affect representative performance?** We show in Figure 6 how scene representative performance deteriorates depending on cycle count and the addition of light cycles, respectively. First, when increasing the number of cycles while maintaining the same distribution of clutter objects across tiers L1, L3, and L4, one can see that the performance of all models deteriorates at a similar rate. Second, when comparing the average accuracy of the VLMs with (L5) and without (L1, L3, L4) the addition of light cycles, performance drops are not as uniform or as strong.

### 5.3 Scene Understanding

In this task, we evaluate the video captioning capabilities of VLMs. Our goal is twofold: (1) to assess whether models can correctly identify all objects and their properties, and (2) to determine whether they can recognize the cyclic transitions introduced in CycliST. The overall evaluation pipeline is illustrated in Figure 5. As in VQA, the model is shown the full video and prompted to describe the scene, including object motions and attribute changes.

Since experiments with a complete evaluation by the LLM-judge (performing both JSON and object matching) failed, we decided on the following two-step process. First, we provide the judge model (Llama3-70B) with the generated caption. Next, the judge maps each object in the caption to a JSON object. In the second stage, we parse the JSON and match each predicted object to its corresponding ground-truth (GT) object. Matching is based solely on object attributes, not on their cyclic transitions, and requires exact agreement on the predicted attributes.

Attributes do not need to be exhaustive for a match; for example, a "red object" can be matched to a "small red rubber cone" in the GT. Each predicted GT object can be matched at most once. Afterward, we compute precision, recall, and F1-score.

Table 11 shows that all models achieve strong performance on object-level scene understanding, with F1 scores between 75% and 90%. Precision and recall are balanced across models. This outcome is unsurprising, as many VLMs have likely been trained on CLEVR (Johnson et al., 2017) or related synthetic datasets with similar object-centric structures. To assess how well models capture cyclic transitions, we further evaluate the correctness of the predicted cycles for each matched object. For each successfully matched object pair, we compute precision and recall by comparing the predicted cycles with the corresponding ground-truth. Cycles associated with unmatched GT objects contribute false negatives, while cycles from unmatched predicted objects count as false positives.

Table 12 shows that there remains a technical gap in VLMs' understanding of dynamic scenes despite strong object-level performance. All models achieve cycle-level F1-scores below 31.2%. Precision and recall show slightly larger variability than in the object-level evaluation, but no consistent trend emerges.

## 6 Related Work

### 6.1 Video Question Answering Datasets

VQA has emerged as a central problem in vision–language research, requiring models to jointly interpret visual dynamics and linguistic queries. To foster progress, numerous datasets have been introduced, spanning real-world and synthetic domains. Real-world benchmarks such as TGIF-QA (Jang et al., 2019), TVQA (Lei et al., 2018), and MovieQA (Tapaswi et al., 2016) provide large collections of videos paired with human-authored questions, targeting models' abilities to comprehend natural language queries grounded in complex visual narratives. Analogously to CycliST, prior real-world datasets have considered estimating the periodicity in videos, e.g., across everyday life scenarios (Panagiotakis et al., 2018; Dwibedi et al., 2020) or in sports (Hu et al., 2022). While grounded in the real world, they are largely limited to single-action analysis rather than incorporating simultaneous or entangled event cycles.

To address this, synthetic diagnostic datasets aim to allow precise control over scene composition and temporal dynamics. MarioQA (Mun et al., 2017) leverages gameplay videos to study temporal dependencies, CLEVRER (Yi et al., 2020) emphasizes causal reasoning with compositional functional programs, and CATER (Girdhar and Ramanan, 2020) probes spatiotemporal reasoning and object permanence.

More recently, Kubric (Greff et al., 2022) introduced a simulator for generating photorealistic videos with fine-grained annotations. Despite these advances, existing datasets primarily capture linear or causal temporal structures and rarely include periodic or cyclical transitions, which are ubiquitous in physical and natural processes. For instance, although CLEVRER considers temporal relations, it does not model cyclic state changes.

CycliST builds upon the diagnostic tradition of CLEVR (Johnson et al., 2017), expanding towards spatiotemporal reasoning. By systematically varying the number of cyclic objects, scene clutter, and illumination, CycliST offers a controlled yet challenging benchmark to evaluate whether current VQA models can detect, track, and exploit cyclical patterns.

## 6.2 Video Language Models

We evaluate Video Language Models for their ability to observe image sequences and answer textual questions about the scene and its evolution. First, we consider LLaVA-OneVision (LLaVA-OV), a large multimodal model designed to unify single-image, multi-image, and video understanding within a single architecture, using the SigLIP vision encoder (Zhai et al., 2023) and the Qwen2 language model (Yang et al., 2024) as its backbone. LLaVA-OV was finetuned on a mixture of single and multi-image data, as well as video data obtained from existing or synthetically annotated data.

Second, we evaluate LLaVA-Video (Zhang et al., 2025) on CycliST, using the single-image checkpoint (no video or multi-image training) of LLaVA-OV, inheriting its encoder and backbone. The model was trained on 178k videos processed with a novel hierarchical frame captioning scheme that groups and summarizes descriptions into three distinct levels. These resulting captions were then used to automatically generate a corpus of multiple-choice and free-form questions. While it kept the frame resolution and frequency of its predecessor, the number of frames is capped to manage GPU memory constraints. Furthermore, it enriches its context window with metadata such as the total frame count and video length. Thirdly, we consider InternVL3 (Chen et al., 2024), which consolidates language and multimodal pretraining into a single stage, rather than training the vision and language modules separately and then training an adapter to connect them. During training, 8-32 frames are sampled from each video, and frames are resized to $448 \times 448$ pixels. InternVideo2.5 (Wang et al., 2025) is another VLM from the Intern family, designed to support longer videos. It is built on the predecessor InternVL2.5. For shorter videos, a dense sampling scheme with 15 frames per second is employed, while for longer videos, a frame rate of 1 frame per second is used. Furthermore, similar tokens are merged using a hierarchical compression scheme that leverages the redundancies of the visual tokens.

As representatives of state-of-the-art proprietary models, we employed Gemini Flash 2.0 and Gemini Flash 2.5 (Anil et al., 2023; Comanici et al., 2025). It was trained on a corpus comprising multiple modalities, including images and video data. In Comanici et al. (2025), Gemini 2.5 was tested with up to 7200 frames. While all of these models demonstrate impressive progress on standard video benchmarks, our experiments reveal that they struggle to generalize to the challenges posed by our work. This underscores a fundamental gap: current architectures are not explicitly equipped to capture periodic dynamics, motivating the need for benchmarks like CycliST that expose this limitation and encourage the development of models with more robust temporal reasoning capabilities.

## 7 Conclusion

**Summary.** We introduce CycliST, a novel benchmark dataset designed to evaluate the reasoning capabilities of Vision Language Models (VLM) on cyclical state transitions. CycliST captures a core aspect of many everyday real-world processes by generating synthetic but richly structured sequences that require models to recognize and reason over periodically emerging patterns. We evaluated the performance of current state-of-the-art (SOTA) Video Question Answering (VQA) models on CycliST and found that, despite their success in prior benchmarks, they struggle to capture and understand periodic patterns, such as linear or orbital motion, and changes in object attributes, including time-dependent color and scale.

Our experiments demonstrate that current SOTA VLMs are incapable of reliably solving tasks that involve detecting and leveraging cyclic temporal dependencies. By highlighting this gap and providing a challenge to measure future computer vision models, CycliST paves the way for the development of more capable agents that can understand repetition not just to make accurate predictions, but to save energy, act efficiently, and decide when to seek new information. Developing these capabilities is crucial for intelligent agents navigating dynamic, temporally structured environments.

**Limitations and Future Work.** While CycliST serves as a useful starting point for exploring cyclical reasoning, it has some limitations to be addressed in the future. First, CycliST is entirely synthetic, meaning it lacks some of the nuances found in real-world data. Future iterations of this work should explore how cyclical structures manifest in real-world scenarios and appropriately expand the benchmark on such scenes. Similarly, real-world cyclicity often involves non-stationary frequencies: the intervals between state transitions are not fixed, and environmental dynamics may vary over time, e.g., in day-night cycles. Moreover, continuous changes in material properties and object positions, along with occasional discrete jumps, are typical of natural phenomena but not yet represented in CycliST. Regarding the VLMs under test, our experiments are deliberately designed to demonstrate the overall performance gap at the benchmark level rather than to reverse-engineer the internal failure modes of each individual model. Our results provide a meaningful step in this direction: models perform well on static scene captioning regardless of cycle count, but fail dramatically when reasoning about cyclic transitions is required. This contrast strongly suggests that the bottleneck lies in temporal and cyclical reasoning rather than basic perception. Nevertheless, a deeper architectural analysis of each VLM's internals would constitute a separate and substantial research contribution. This includes thorough ablations of how individual scene attributes, such as clutter, lighting changes, and cyclical patterns, relate to individual architecture's failure modes. Finally, CycliST omits causal events as found in real-world scenarios, e.g., the effects of a traffic light on the flow of vehicles at an intersection. Including such dependencies while maintaining system stability and avoiding chaotic movements that disrupt the scene's cycling remains as future work.

## Broader Impact Statement

This work introduces CycliST, a benchmark dataset designed to advance the capabilities of Video Language Models in understanding cyclical state transitions. The primary positive impact of our contribution is providing the research community with a controlled, synthetic environment for evaluating spatio-temporal reasoning on image data, which can lead to more capable and reliable models in fields such as robotics, education, and accessibility. Improving the fundamental visual perception and reasoning of AI models is a critical step towards building more robust and safer systems. This focus on core competencies mitigates the risks associated with unpredictable model behavior and promotes more generalizable and reliable AI. However, one must be conscious of the adverse tasks enabled by increasingly capable machine learning models. By making CycliST publicly available under a permissive license, we aim to encourage transparent, reproducible, and responsible innovation.

## Acknowledgments and Disclosure of Funding

Simon Kohaut gratefully acknowledges financial support from the Honda Research Institute Europe (HRI-EU). Daniel Ochs gratefully acknowledges the financial support from the EU project EXPLAIN, under the BMFTR (grant 01-S22030D). This work was supported by the Konrad Zuse School of Excellence in Learning and Intelligent Systems (ELIZA). We gratefully acknowledge support from the hessian.AI Service Center (funded by the Federal Ministry of Research, Technology and Space, BMFTR, grant no. 16IS22091) and the hessian.AI Innovation Lab (funded by the Hessian Ministry for Digital Strategy and Innovation, grant no. S-DIW04/0013/003). Further we acknowledge support from hessian.AI Connectom Networking and Innovation Fund as part of the project "Explainable AI for Efficient DNN Inference on Hardware" via hessian.AI. The TU Eindhoven authors received support from their Department of Mathematics and Computer Science and the Eindhoven Artificial Intelligence Systems Institute.

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

# Appendix A.

Table 13: CycliST's functions for template-based question-generation.

| Group | Name and Description | Input Type | Output type |
|---|---|---|---|
| **Scene** | `scene`
Returns all objects in the scene | | [objects] |
| | `unique`
Selects a unique object, returns invalid otherwise | [objects] | object |
| **Filter** | `filter_color_⟨quantifier⟩`
Selects objects based on color and quantifier | ([objects], value) | [objects] |
| | `filter_shape`
Selects objects with a specific shape | ([objects], value) | [objects] |
| | `filter_size_⟨quantifier⟩`
Selects objects based on size and temporal quantifier | ([objects], value) | [objects] |
| | `filter_material`
Selects objects with a specific material | ([objects], value) | [objects] |
| | `filter_orbit`
Selects objects that are orbiting | ([objects], value) | [objects] |
| **Query** | `query_color`
Returns the color of an object. Returns invalid if color changing. | object | value |
| | `query_color_⟨quantifier⟩`
Returns the initial or final color of an object | object | value |
| | `query_size`
Returns the size of an object. Returns invalid if resizing. | object | value |
| | `query_size_⟨quantifier⟩`
Returns the initial or final size of an object | object | value |
| | `query_shape`
Returns the shape of an object | object | value |
| | `query_material`
Returns the material of an object | object | value |
| | `query_orbit_direction`
Returns the orbit direction (clockwise / counterclockwise) | object | value |
| | `query_⟨cycle⟩_period`
Returns the frame duration of the cyclic object | object | value |

Table 13: CycliST's functions for template-based question-generation.

| Group | Name and Description | Input Type | Output type |
|---|---|---|---|
| | query_⟨cycle⟩_passes
Returns the number of cycles the object completes within the video | object | value |
| Logic | logical_or
Performs a logical OR operation | (value, value) | value |
| | logical_and
Performs a logical AND operation | (value, value) | value |
| | logical_not
Performs a logical NOT operation | value | value |
| Relate | relate_⟨quantifier⟩
Returns all objects that relate to the input object | (object, value) | [objects] |
| Sets | union
Returns the union of two object sets | ([objects], [objects]) | [objects] |
| | except
Returns the set difference of two object sets | ([objects], [objects]) | [objects] |
| | intersect
Returns the intersection of two object sets | ([objects], [objects]) | [objects] |
| | include
Checks if an object is in an object set | (object, [objects]) | value |
| | exist
Checks if any object exists in a set | [objects] | value |
| | count
Returns the number of objects in a set | [objects] | value |
| Same | same_size
Returns all objects with the same size | object | [objects] |
| | same_color
Returns all objects with the same (current) color | object | [objects] |
| | same_material
Returns all objects with the same material | object | [objects] |
| | same_shape
Returns all objects with the same shape | object | [objects] |
| Compare | equal_color
Checks if two colors are equal | (value, value) | value |
| | equal_color_⟨quantifier⟩ | (object, object) | value |

Table 13: CycliST's functions for template-based question-generation.

| Group | Name and Description | Input Type | Output type |
|---|---|---|---|
| | Checks if two objects have equal color based on quantifier | | |
| | `equal_size` 
 Checks if two sizes are equal | (value, value) | value |
| | `equal_size_⟨quantifier⟩` 
 Checks if two objects have equal size based on quantifier | (object, object) | value |
| | `equal_shape` 
 Checks if two shapes are equal | (value, value) | value |
| | `equal_material` 
 Checks if two materials are equal | (value, value) | value |
| | `equal_integer` 
 Checks if two integers are equal | (value, value) | value |
| | `less_than` 
 Checks if the first integer is less than the second | (value, value) | value |
| | `greater_than` 
 Checks if the first integer is greater than the second | (value, value) | value |
| | `equal_object` 
 Checks if two objects are the same | (object, object) | value |

## Appendix B. Question Statistics

Table 14: Number of Questions per Template per Dataset on the test split

| CycliST Tier Question Template | Unicycle | Unicycle-Cl. | Bicycle | Tricycle | Nightrider | Total |
|---|---|---|---|---|---|---|
| descriptive_existential_attributes | 750 | 750 | 750 | 769 | 712 | 3731 |
| descriptive_existential_compare | 726 | 748 | 736 | 765 | 702 | 3677 |
| descriptive_existential_relate | 745 | 750 | 750 | 769 | 711 | 3725 |
| descriptive_universal_attributes | 750 | 750 | 750 | 769 | 712 | 3731 |
| descriptive_universal_compare | 738 | 748 | 747 | 766 | 705 | 3704 |
| descriptive_universal_relate | 745 | 750 | 748 | 767 | 710 | 3720 |
| cycle_representative_orbit | 148 | 147 | 249 | 325 | 260 | 1129 |
| cycle_representative_clockwise | 135 | 128 | 225 | 308 | 240 | 1036 |
| cycle_representative_transition | 279 | 283 | 427 | 539 | 404 | 1932 |
| numeric_counting | 312 | 312 | 468 | 551 | 502 | 2145 |
| numeric_periodicity | 177 | 185 | 195 | 204 | 199 | 960 |
| numeric_occurrence | 125 | 133 | 143 | 155 | 147 | 703 |
| **Total** | 5630 | 5684 | 6188 | 6687 | 6004 | 30193 |

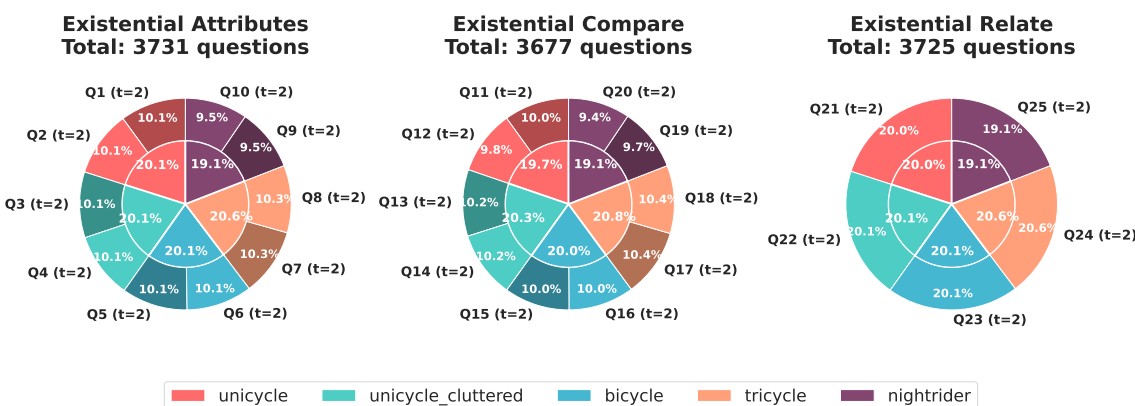

Figure 7: Answer Distribution Temporal Descriptive Existential

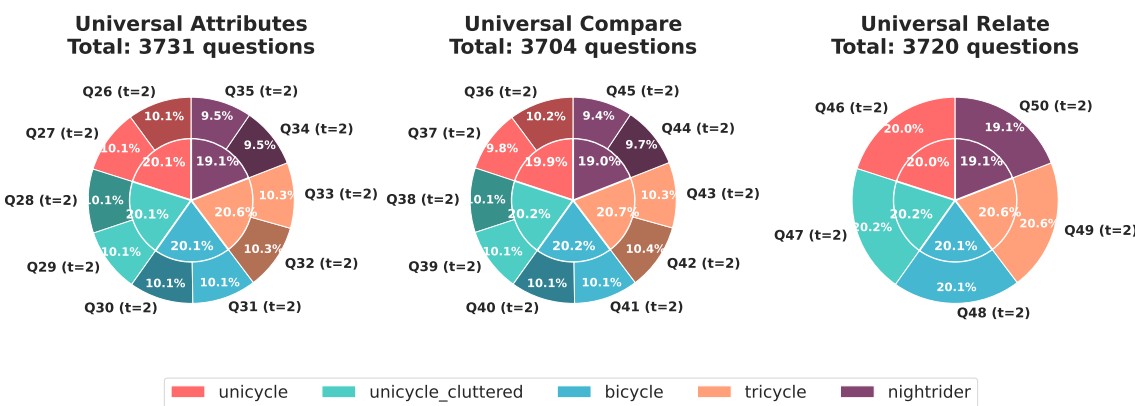

Figure 8: Answer Distribution Temporal Descriptive Universal

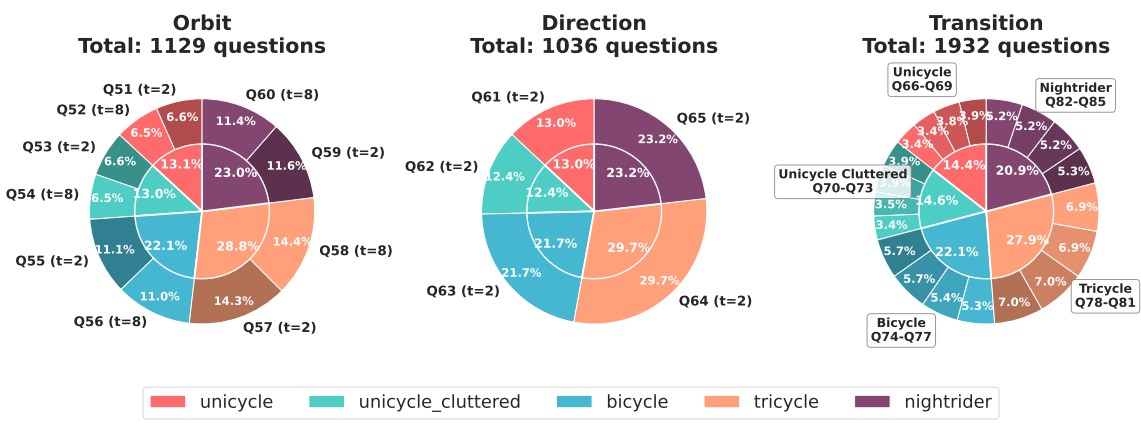

Figure 9: Answer Distribution Scene Representative Cyclic

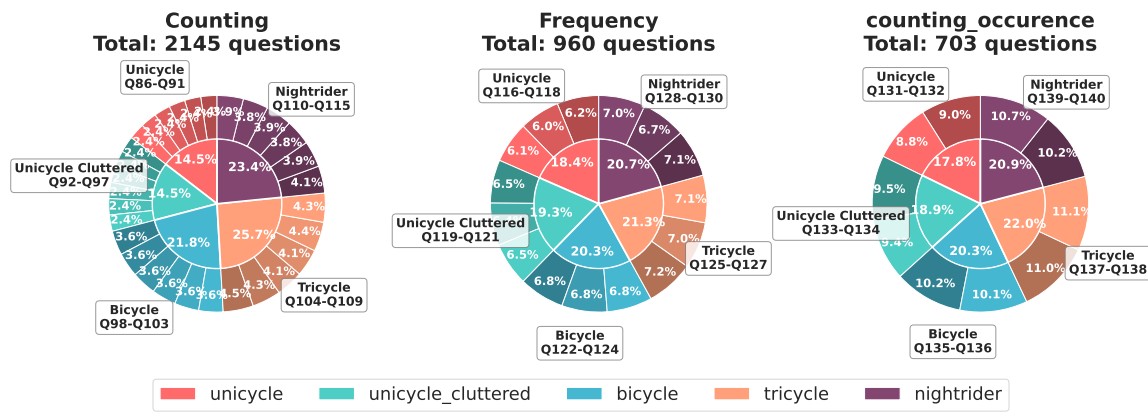

Figure 10: Answer Distribution Scene Representative Cyclic

