# OpenReview forum: "CycliST: A Video Language Model Benchmark for Reasoning on Cyclical State Transitions"
_DMLR — Accepted by DMLR_

### Review · Reviewer_FQNU · 2026-03-23

**Recommendation:** 4
**Confidence:** 2

**Summary Of Contributions:**

1. Identifies an underexplored problem: current video-language models show clear limitations in handling dynamic scenarios.

2. Proposes a new benchmark (CycliST) to evaluate such capabilities in a controlled setting.

3. Frames these limitations under a unified perspective, highlighting temporal and cyclical reasoning as key weaknesses of existing models.

**Strengths:**

1. The contribution is clear and focuses on a relatively underexplored aspect. The paper introduces CycliST, a novel benchmark explicitly targeting cyclical state transitions in videos, which is a largely overlooked dimension in video understanding. This fills an important gap in prior work, which mainly focuses on linear or causal temporal reasoning rather than periodic dynamics.

2. The paper is built upon prior datasets rather than being entirely from scratch, which makes it more beneficial for the community. It extends CLEVR-style datasets from static settings to dynamic scenarios, making it easier for the community to adopt and build upon, and promoting cumulative progress.

3. The experiments clearly show that current models perform poorly on this type of capability. This provides a clear direction for future research and highlights an important area for improvement in video-language models.

4. The dataset is large-scale, with strong data coverage and good reproducibility, which increases its practical value for benchmarking and future research.

**Audience:**

Yes

**Broader Impact Concerns:**

No concerns

**Claims And Evidence:**

The empirical results show that models perform poorly on the proposed tasks. This part is supported.

However, these claims depend on the validity of the dataset design. The tiered setup (L1–L5) is not clearly justified. Multiple factors change at the same time (e.g., number of objects, interactions, clutter, lighting), so it is unclear what actually causes the performance drop.

Because of this, the results show that models fail on the dataset, but do not clearly explain why. Either the design needs to be better justified, or more controlled ablation studies are needed.

Overall, the evidence supports the observed performance gaps, but is not sufficient to fully support stronger conclusions about model capabilities.

**Datasets And Benchmarks:**

Yes the paper provides a clear description of the dataset generation process and includes sufficient details to support reproducibility. The use of a synthetic pipeline, along with the release of code and data, makes the benchmark relatively easy to reproduce and extend.

Only problem there is limited discussion on intended use, limitations, and long-term maintenance.

**Extended Submissions:**

Yes it meets.

**Limitations:**

1. The theoretical analysis behind the L1–L5 evaluation framework appears to be insufficient. It is unclear whether the difficulty across tiers is measurable or disentangled in a principled way.

2. The current analysis can only conclude that “higher tiers are more difficult,” but it cannot precisely identify which factors contribute to the increased difficulty. Each tier often modifies multiple sources of complexity simultaneously. The transition from L3 to L4 does not only increase the number of objects, but also introduces more complex cycle interactions and phase relationships. It is unclear whether the observed performance degradation can be attributed to any specific factor.

3. The dataset is entirely synthetic and lacks real-world video data. The generalizability of the conclusions remains unclear. It would be helpful to include real-world scenarios, provide more detailed case studies, or explain the rationale for not incorporating real-world data.

4. The evaluation for some question types relies heavily on an LLM-as-a-judge. For these open-ended tasks, it would be helpful to introduce a multiple-choice (MCQ) evaluation as a complementary or comparative setting, in order to validate the reliability of the LLM-based evaluation pipeline.

**Requested Changes:**

- **Evaluation (LLM-as-a-judge)**: reduce the reliance on LLM-based judging, or at least validate it more carefully. For example, add a controlled evaluation setting (e.g., MCQ) to compare with the current pipeline and show consistency.

- **Tier design (L1–L5)**: provide a clearer justification of the tiered design. Ideally, isolate different factors (e.g., number of cycles, clutter, lighting) and analyze their individual contributions to difficulty, instead of changing multiple factors at once.  Or add some theoretical analysis.

- **Analysis depth**: go beyond reporting accuracy and provide more diagnostic analysis. For example, distinguish whether failures come from perception, object tracking, or temporal reasoning, and include more fine-grained breakdowns or case studies.

- **Real-world relevance**: either include some real-world examples / evaluation, or more explicitly discuss why a fully synthetic setup is sufficient and what limitations it brings in terms of generalization.

**Strengths And Weaknesses:**

- **Dataset**: introduces a relatively new aspect (cyclical state transitions), which is interesting and not well covered in prior work. The scale is also decent and useful for benchmarking. But everything is synthetic, so it is unclear how well the conclusions transfer to real-world scenarios.

- **Design (L1–L5)**: the tiered setup is intuitive and makes the benchmark easy to understand. However, multiple factors change at the same time across tiers (e.g., number of objects, interactions, phase relations), so it is hard to tell which factor actually drives the difficulty.

- **Related work**: nicely extends CLEVR datasets from static to dynamic settings, which is helpful for the community. But this also means the setup may inherit some limitations of synthetic environments.

- **Empirical findings**: clearly shows that current models struggle with this type of reasoning, which is valuable and gives a clear research direction. At the same time, the analysis is mostly descriptive and does not really explain why models fail (e.g., perception vs. tracking vs. reasoning).

- **Evaluation**: covers a range of tasks and metrics, which is good. But some parts rely heavily on LLM-as-a-judge, which may introduce bias. It might be better to include a more controlled setup (e.g., MCQ) as a comparison.

---

### Review · Reviewer_BJKV · 2026-03-25

**Recommendation:** 4
**Confidence:** 2

**Summary Of Contributions:**

This paper introduces CycliST, a novel synthetic video benchmark dataset designed to evaluate the spatiotemporal reasoning capabilities of Video Language Models, with a specific focus on cyclical state transitions. The dataset contains 14.8k generated videos and approximately 120k question-answer pairs featuring repeating physical patterns, such as linear back-and-forth motion, orbital motion, and periodic changes in visual attributes.

The authors innovatively use formal logic quantifiers (Universal "always" and Existential "ever") to generate questions that rigorously test temporal consistency. By testing a range of state-of-the-art VLMs across a 5-tier difficulty system such as L1 to L5, the authors reveal a critical finding: current VLMs struggle severely with periodic dynamics. The models perform at near random-guessing levels for directional tracking and fail almost entirely at counting cycle occurrences, exposing a significant architectural blind spot in the community.

**Strengths:**

S1. Significance & Novelty: Periodic phenomena (e.g., traffic lights, mechanical gears, biological rhythms) are fundamental in the physical world. While existing video benchmarks heavily overfit to linear or single-shot actions (like collisions or picking up objects), CycliST takes a pioneering step into continuous periodic dynamics. Exposing the fact that modern VLMs are essentially "period-blind" is a deep and valuable insight for the broader research community.

S2. Rigorous Methodological Design: The integration of Universal ("always") and Existential ("ever") quantifiers into the VQA generation is an elegant and mathematically sound way to test true temporal consistency across the entire video timeline.

S3. Excellent Tiered Evaluation: The progressive difficulty design allows for surgical diagnosis of exactly what visual complexities cause model performance to degrade.

**Audience:**

Yes

**Broader Impact Concerns:**

The authors have included a satisfactory Broader Impact Statement. Because the dataset relies entirely on procedurally generated synthetic geometric objects, it carries minimal risk regarding privacy, toxicity, or harmful historical biases typically found in web-scraped datasets. The goal of improving spatiotemporal reasoning is generally beneficial for safety-critical systems like autonomous driving. I have no further ethical concerns.

**Claims And Evidence:**

Most claims are well-supported by clear evidence, particularly the claim that VLMs struggle with tracking cyclical patterns (supported by near-random accuracy in Tables 6, 7, and 8 across models from 7B to 78B parameters).
However, two claims currently lack fair/sufficient evidence:

The claim that VLMs "have no notion of frames per second" (Section 5.2, Q7) is misleading. The evidence (MAE ~100 frames) does not cleanly support this claim because the experiment design ignores the confounding variable of VLM input preprocessing (sparse frame sampling). A model cannot count frames it was never fed.

The claim regarding the massive drop in cycle-level scene understanding (Table 12) lacks sufficient robustness evidence (human-validation) to rule out automated grading errors caused by strict JSON matching.

**Datasets And Benchmarks:**

Yes, there is sufficient detail. The authors have provided anonymized links to HuggingFace and GitHub. The details on scene validation, Blender rendering, and split distributions are thorough and sufficient to support full reproducibility.

**Extended Submissions:**

N/A.

**Limitations:**

The authors adequately discuss the limitations of their dataset in the Conclusion, specifically noting that it is entirely synthetic, lacks non-stationary frequencies (e.g., varying intervals), and omits causal physical events. However, they failed to discuss the methodological limitations of their own evaluation pipeline—specifically, the impact of VLM sparse frame subsampling on time-based queries, and the potential false-negative rate of strict automated LLM-as-a-judge JSON matching. Addressing these in the revision is necessary.

**Requested Changes:**

Critical (Required for securing my recommendation for acceptance):

1. Address the unfair "Frame Counting" Task (Section 5.2 / Table 10): Modern VLMs utilize "sparse frame sampling" (e.g., uniformly extracting only 8 to 32 frames from a video) to manage context limits and memory. They physically do not receive all 160 frames or the absolute FPS metadata. Asking them to predict the exact absolute physical frame count is an unfair test of their architecture. You must either: (A) remove this specific task; (B) change it to a relative time metric (e.g., "What fraction of the video does one cycle take?"); or (C) explicitly acknowledge in the text that this massive error is an expected architectural artifact of sparse sampling, rather than a pure cognitive failure ("having no notion of time").

2. Validate the LLM-Judge Pipeline (Section 5.3 / Table 12): The extremely low F1 scores (<31%) on cyclic transitions might simply be a result of the LLM-judge failing to parse diverse natural language into the strict JSON format required for "exact agreement". You must conduct a human evaluation on a random sample (e.g., 100-200 VLM generated answers marked as incorrect by the script) and report the "Human-LLM Agreement Rate". This will prove whether these low scores represent genuine VLM visual failures, or just a brittle grading script.

To strengthen the work:

3. Fix Table 4 Headers: The crucial temporal quantifiers "Universal" and "Existential" are completely missing from the column headers of Table 4. Please add multi-level headers so readers can actually understand what the top and bottom halves of the table represent.

4. Add Visual Baselines to Tables: For binary classification tasks (like Table 6 for orbit direction, where 50% is random guessing), please add a visual baseline indicator (e.g., a shaded row or a bold line at 50%). This will make the severity of the models' failures instantly obvious to the reader.

**Strengths And Weaknesses:**

Strengths:

The paper identifies a foundational and largely overlooked problem in video understanding: cyclical reasoning. The dataset generation is mathematically rigorous, and the tiered evaluation framework serves as an excellent diagnostic tool. Integrating formal logic quantifiers into the VQA pipeline is a highly effective way to prevent models from taking single-frame shortcuts.

Weaknesses:

There is a critical flaw in the experimental design regarding the "absolute frame counting" task, which unfairly penalizes models for their standard preprocessing architectures. Additionally, the automated evaluation pipeline for the Scene Understanding task relies on a rigid "exact agreement" JSON matching by an LLM-as-a-judge, which likely produces a high rate of false negatives that misrepresent the models' true capabilities. Lastly, the presentation of key results (e.g., Table 4) lacks crucial text headers and visual baselines.

---

### Review · Reviewer_wWYy · 2026-04-08

**Recommendation:** 4
**Confidence:** 2

**Summary Of Contributions:**

This work introduces CycliST, a novel high-resolution, high-framerate benchmark dataset specifically designed to evaluate the reasoning capabilities of state-of-the-art Video Language Models (VLMs) on cyclical state transitions in video sequences.

**Strengths:**

1. The benchmark extends prior synthetic diagnostic frameworks by integrating diverse cyclical dynamics, including linear and orbital motion cycles, periodic attribute changes (size, color, orientation), and scene-wide light intensity cycles, with physically based rendering for photorealistic, temporally consistent video content.

2.CycliST provides a comprehensive, tiered evaluation protocol with 5 difficulty levels that scale complexity via the number of cyclic objects, scene clutter, and lighting variations, and comprises 14,800 full-HD videos paired with 120,000 annotated question-answer pairs for systematic and fine-grained VLM assessment.

3. Through extensive experiments on leading open-source and proprietary VLMs, this work reveals critical limitations of current models in generalizing to cyclical dynamics, demonstrating that existing VLMs lack robust temporal understanding, struggle with quantitative cyclic reasoning, and show no consistent performance correlation with model size or architecture.

**Audience:**

Yes

**Claims And Evidence:**

Yes

**Datasets And Benchmarks:**

Yes

**Extended Submissions:**

No

**Requested Changes:**

See weakness 2,3

**Strengths And Weaknesses:**

1. The scene generation based on synthetic geometric primitives rendered via Blender, the template-based functional question-answer pair generation, and the evaluation paradigm combining Video Question Answering (VQA) and scene description are entirely inherited from the well-established framework of the CLEVR series. The only marginal additions are cyclic motion/attribute changes and temporal quantifiers.

2. The paper omits a substantial body of foundational and state-of-the-art work on benchmark datasets and modeling methods related to cyclic temporal reasoning, periodic event localization in videos, and repetitive action recognition[1],[2].

3.The selection of evaluated Video Language Models (VLMs) is excessively narrow. The authors should supplement the evaluation results for the Omini series models to expand the breadth and generalizability of the experimental conclusions.

[1] TransRAC: Encoding Multi-scale Temporal Correlation with Transformers for Repetitive Action Counting.
[2] Counting Out Time: Class Agnostic Video Repetition Counting in the Wild